# Comparison of Grip Strength in Recreational Climbers and Non-Climbing Athletes—A Cross-Sectional Study

**DOI:** 10.3390/ijerph18010129

**Published:** 2020-12-27

**Authors:** Mara Assmann, Gino Steinmetz, Arndt Friedrich Schilling, Dominik Saul

**Affiliations:** 1Department of Trauma, Orthopedics and Reconstructive Surgery, Georg-August-University of Goettingen, 37075 Goettingen, Germany; Mara.assmann@t-online.de (M.A.); Gino.Steinmetz@med.uni-goettingen.de (G.S.); Arndt.schilling@med.uni-goettingen.de (A.F.S.); 2Kogod Center on Aging and Division of Endocrinology, Mayo Clinic, Rochester, MN 55905, USA

**Keywords:** climbing, indoor climbing, grip strength, upper extremity measurements, handedness

## Abstract

In recent years, climbing sports is on the rise making its Olympic debut in 2021. Physiological traits of professional rock climbers have been intensively studied, while recreational indoor climbers are less investigated, especially regarding grip strength and upper extremity proportions. In this cross-sectional study, we aimed to understand what discerns the recreational climber from disparate recreational athletes. Therefore, we analyzed 50 recreational climbing (30.3 ± 12.7 years, 1.76 ± 0.09 m and 67.0 ± 14.0 kg) and 50 non-climbing athletes (26.4 ± 9.1 years, 1.78 ± 0.09 m and 73.2 ± 12.6 kg) to detect differences in their finger grip strength of seven different pinches. In addition, the upper extremity proportions were measured. Even in recreational climbers, almost all analyzed grips were stronger compared to other athletes (*p* < 0.05 in all but non-dominant fist, small to moderate effect sizes). Only the grip strength of the whole non-dominant hand was not significantly different (*p* = 0.17). Interestingly, differences between the dominant and non-dominant hand appeared to be larger in the non-climbing (all *p* < 0.05, all but one with small effect size) compared to the climbing cohort (pinch I/IV and pinch I/II+III+IV not different and mostly trivial). Circumference measurements showed that 10 cm below the lateral epicondyle, climbers exhibited significantly greater perimeter compared to non-climbing athletes (*p* < 0.05, small effect size). Our results show that recreational climbers have higher measured grip strength, but less profound differences between the dominant and non-dominant hand.

## 1. Introduction

For the climbing sport, the rise in popularity came with a rise in the number of recreational climbing athletes [1,2,3,4]. Throughout the literature, highly advanced climbers, the elite of their profession, are well-characterized. Comparing elite climbers with advanced climbers, a superior finger grip strength of the dominant hand has been demonstrated, as has enhanced muscular endurance [5,6,7]. In addition, elite climbers have been proven to have a slower breathing frequency, larger muscle oxygenation and well-developed limb musculature compared to non-climbers [8,9]. Several years of climbing lead to a muscular adoption with functional prevalence of fast resistant motor units in very advanced climbing athletes and a specific physical profile, defined by high shoulder power as well as finger, hand and arm strength [6,10,11].

These studies characterize mainly elite and very advanced climbing athletes, while there is a lack of studies focusing on recreational climbers [1]. The trend of climbing sport spreading more and more into a leisure sports activity, however, punctuates the importance of focusing on this particular cohort, which might find itself underrepresented in current studies [12].

It was demonstrated in the elite climbing population, that the individual grip strength and endurance is associated with the climbing ability. To which extend this is also true for recreational climbers, has not been demonstrated so far since climbing athletes, which have not (yet) reached the top level, but practice on a regular basis, have not been described abundantly [12,13,14,15]. Whether these climbers can be discerned from regular sportsmen with regards to grip strength and physical properties in a similar manner that elite climbers can, is not known [1].

These insights might help amateurs to specifically focus their training in order to catch up deficits compared to advanced climbers. With additional knowledge on what distinguishes recreational climbers from comparable athletes of a different sport, the characteristics might become obvious, with which a climbing career is about to start. After these are reached, the specific traits of elite climbers might be the next step to practice for.

In order to understand the physical traits of recreational climbers, we aimed to characterize the casual climber apart from other athletes. The objective of our study was to determine which particular physical (grip force and circumference of upper and lower arm) characteristics discern the casual climber from comparably active athletes. This might serve as a blueprint for amateurs, who want to start a climbing career. Anticipating certain physical and physiological characteristics from previous studies of elite climbers [5,6,11,13,16,17], we sought to identify, whether these characteristics were already detectable in recreational climbers compared to similarly active sportsmen in order to understand, which characteristics are the first to pursue when starting with the climbing sport.

## 2. Materials & Methods

### 2.1. Experimental Approach to the Problem

We aimed to conceive if certain characteristics that are well-described in elite climbers, can be similarly assessed even in recreational climbers. Which of the superior characteristics defining the best climbers are already detectable on a recreational level and, subsequently, the first ones to reach when an athlete starts with the climbing sport?

To assess these very initial characteristics, we assumed that athletic human beings are the most probable to start with the climbing sport [18]. Accordingly, we recruited the comparison group from active athletes via announcement at the local university with no experience in climbing or related sports. We chose an active control group to exclude measuring just general training-related effects. These would have interfered when a non-active control group would have been chosen. The climbing group was recruited from recreational climbers at a local climbing center with an intermediate or advanced climbing level.

Taking into account that hand grip-strength and upper arm circumference parameters are widely studied in the elite climbing community indicating superior strength as arm extent in top athletes [1,6,11,19,20,21], we chose these parameters to assess whether this could as well be found in recreational climbers. In particular, we felt that athletes performing fitness and ball sports might also show proper overall grip strength, allowing for a better differentiation for what is specifically attributable to the recreational climbing athlete. To identify which of the aspects that were seen in top climbers are similarly pronounced in just recreational climbers, we assessed both grip strength in the dominant and non-dominant hand as well as upper extremity circumferences in advanced climbers and compared these to athletic controls. This should help beginning climbing enthusiasts to better understand what the typical physical requirements are for this unique sport.

### 2.2. Subjects

For a power of 0.95 (estimated effect size ρ = 0.5, α = 0.05), the necessary sample size was calculated to be *n* = 42. Accounting for possible losses, we cross-sectionally analyzed 50 climbing and 50 non-climbing athletes (40% fitness, 20% running, 18% ball sports, 22% others) from 09/18 to 12/19 and assessed their grip strength regarding seven different pinch grips for both the non-dominant and dominant hand. In both groups, age, gender, height, weight, body mass index (BMI) and training years were assessed (Supplementary Appendix A). Additionally, in the former group, the climbing grade (International Union of Alpine Associations UIAA [UIAA] and International Rock Climbing Research Association [IRCRA]), years of climbing and training frequency was assessed. The study was approved by the local ethics committee (11/7/18) and all participants gave informed consent.

### 2.3. Inclusion Criteria

For the climbing group, only athletes were considered who were regular climbers at the RoXx-climbing center in Goettingen, Germany, or DAV climbing hall Hildesheim, Germany, with an intermediate and advanced climbing level according to Draper et al. [22]. For the non-climbing group, the athletes should perform sport on a regular basis, but no climbing-associated sport like (outdoor) climbing, bouldering or ice climbing. Both cohort characteristics are presented in Appendix A. 

### 2.4. Exclusion Criteria

We excluded athletes under 18 years or with previous surgery or acute injuries in one of the examined arms.

### 2.5. Circumference of Upper and Lower Arm

To indirectly assess muscular strength, nine points on upper and lower arm were measured on each side before grip force examination [23,24]. The assessed points of measurement were: 15 cm above lateral epicondyle, greatest circumference of upper arm, elbow, greatest circumference of lower arm, 10 cm below lateral epicondyle, wrist, metacarpus (without thumb), length of arm (shoulder to distal radius/Processus styloideus radii), and greatest distance in cm from thumb to little finger as well as arm span (according to F4222 of DGUV). Arm span relative to height was auxiliary measured (referred to as ape index [25]).

### 2.6. Measurement of Grip Force

By reason of grip force being highly trained in elite climbers, we aimed to assess whether these effects were measurable in recreational climbers compared to non-climbing athletes as well. We therefore determined grip strength by twice repeated measurements with a pause of half a minute between assessments of grip force using a mechanograph in a free position (sitting on a chair, upper arm leant on thorax, 90° flexed elbow, pronated hand, no compensatory movements tolerated; Leonardo Mechanograph^®^ GF, Novotec Medical GmbH, Pforzheim, Germany, Appendix A). The measurements were made in kilograms (kg). Usage of a dynamometer to assess grip strength has been shown to be reliable elsewhere [15]. To eliminate fatigue effects, studies were based on the mean of both measurements. Fingers were enumerated from thumb (I) to little finger (V) and the “/” distinguished the two sides of the grip. The examined grips were Pinch I/II, Pinch I/III, Pinch I/IV, Pinch I/III + IV, Pinch I/II+III, Pinch I/II+III+IV and Fist (Pinch I/II+III+IV+V). Both the dominant (Dom) and non-dominant (Non-dom) hand were examined, named after the side that was used for writing [26].

### 2.7. Statistics

Analyses were performed with GraphPad Prism 8.02 (GraphPad Software, Inc., San Diego, CA, USA) and SPSS Statistics 26.0 (IBM, Armonk, NY, USA).

The normal distribution of continuous variables was tested by the d’Agostino Pearson-Test. The Student’s unpaired t-test was used if not stated differently. Otherwise, Mann-Whitney test was applied. Categorial variables were analyzed using the Fisher’s exact test. Intraclass correlation was used to determine reliability for test-retest. If not stated differently, ±standard deviation is presented. Depicted values are mean and the 95% confidence interval (CI). Significant differences are marked with asterisks (*** *p* < 0.001, ** *p* < 0.01, * *p* < 0.05).

The effect size (ES) was calculated according to López-Rivera and González-Badillo, Hedges and Olkin and Rhea [27,28,29] with g = (mean pretest)-(mean posttest)/pooled standard deviation of corresponding standard deviations. The pooled standard deviation (SDpool) was calculated with the formula $SDpool=\sqrt{\frac{\left(SD_{1}^{2}+SD_{2}^{2}\right)}{2}}$ [30]. Therefore, g < 0.25 was defined as trivial, 0.25–0.5 as small, 0.5–1 as moderate and >1 as large.

## 3. Results

### 3.1. Characteristics of Climbers and Non-Climbers

We included 50 climbing and 50 non-climbing athletes in this study. The age of the climbing group (30.3 ± 12.7 years) did not differ significantly from the age of the non-climbing cohort (26.4 ± 9.1 years) (*p* = 0.1, Mann-Whitney test) with a mean age (all participants) of 28.3 years. Neither BMI nor gender or handedness differed significantly between the groups (Appendix A). Within the climbing group, the self-reported climbing grade varied between IRCRA levels 10–23 (mean IRCRA 16.88 ± 4.92, mean French: 6c+/7a, UIAA: VI to X-, mean VIII-/VIII). The climbing group had a mean climbing experience of 6.8 (±7.3) years, while the non-climbing group had no climbing experience at all. Ape index did not differ among groups (climbers: 1.013 ± 0.02 and non-climbers 1.010 ± 0.02, *p* = 0.36, unpaired t-test, Appendix A).

### 3.2. Circumference of Upper and Lower Arm

To assess physical differences among cohorts, measurements of circumference on defined locations of upper and lower arm were performed. Analysis revealed only one significant difference which was 10 cm below the lateral epicondyle. On this location, climbers had a significantly greater circumference compared to non-climbers, while the effect size (ES) was small (Table 1).

### 3.3. Hand and Finger Grip Strength

The calculated test-retest reliability of grip strength measurement was 0.9898 for one measurement (95%-CI: 0.9887–0.9908), 0.9998 for the mean (95%-CI: 0.9996–0.9999). The finger grip strength was measured in seven different pinch configurations, all numbered by their involved fingers. After two measurements, the mean values for the resulting force were calculated for both hands. In almost all but the non-dominant fist force, climbers reached higher values compared to athletic non-climbers (Table 2). The highest discrepancy yielded in Pinch I/II as well as Pinch I/IV and Pinch I/II+III+IV, all being moderate in effect size. After adapting the pinch force to BMI, similar results were detected. As well, fist force on the non-dominant side was not significantly different among groups (Appendix A). Force differences among the dominant and non-dominant hand showed an overall superiority of the former in every pinch for climbers as well as non-climbers. Yet the non-climbing cohort showed a significantly greater difference between dominant and non-dominant hand in every pinch, whereas these differences were less apparent in the climbing cohort, especially for unusual grips like Pinch I/IV and Pinch I/II+III+IV. Thus, that differences between the dominant and non-dominant hand were not as strong in climbers compared to non-climbers (Table 3).

## 4. Discussion

With this study, we aimed to assess whether the superior grip strength and upper extremity circumferences which were detected in elite climbers, could similarly be found in advanced climbing athletes. The initial characteristics, which discern common athletes from recreational climbing athletes were to be discovered in order to “blueprint” what beginning climbers should especially practice or which characteristics were prominent in advanced climbing athletes.

In order to understand these differences among climbing and non-climbing athletes, we performed the first study with 100 recreational athletes that were assessed regarding their grip strength and upper extremity measurements, distinguishing the dominant and non-dominant hand. The group and age characteristics of the recreational climbers and non-climbing athletes were comparable to similar studies involving advanced athletes [17,31].

### 4.1. The Circumference of Lower and Upper Arm Shows that Just One Measuring Point Is Enhanced in Recreational Climbers

We detected a difference between climbers and non-climbers in just one of the circumference measurements, indicating a taller upper forearm in climbers compared to non-climbers. Consequently, for amateur climbers, the upper forearm may be considered a primary target of initial training, especially since it has been demonstrated by Usaj et al. that muscle strength-training in climbers could increase performance of forearm muscles substantially [32]. However, the effect size was small and the association of arm circumference with strength can be questioned since there is no linear correlation between them, especially in women [23,24].

Differences among forearm and arm circumference have been measured in another study, although the height of measurement was not stated exactly. In a climbing cohort, which was more experienced than our recreational group (6b to 8c French vs. 6c + /7a), arm circumference was 32.7 cm and significantly 2 cm more than in the untrained group (30.9 cm) [13]. In contrast, our recreational and non-climbing group did not differ significantly in the greatest upper arm circumference (30.81 cm vs. 31 cm). The underlying reason could be that Tomaszewski et al. measured the right side (not necessarily dominant) and under muscular contraction, while we measured the quiescent state. Our forearm measurements, however, were comparable for the untrained cohort (28.3 cm vs. 26.02 cm, while we measured 26.9 cm vs. 26.17 cm). Nonetheless, the differences 10 cm below lateral epicondyle that we detected, can be similarly seen in the study by Tomaszewski et al., indicating that specific body compositions such as forearm volume favor climbing performance [13], which is additionally confirmed in a study by Arazi et al. [4]. The group of Ozimek et al. measured twenty climbing athletes with an IRCRA-level of 25–27 and 22–23, therein comparing elite and advanced climbers. The overall arm circumference was 28.92 cm vs. 30.21 cm, forearm circumference 28.50 cm and 28.63 cm, the latter yet indicating the much higher climbing level compared to our study since the forearm values were much larger. On the upper arm, consistently, the values of “advanced climbers” in their cohort and climbers in our cohort were comparable (30.21 cm and 30.26 cm) although IRCRA-levels were not [6]. Similar to our investigation, these three studies indicate a preferentially enhanced upper forearm circumference in climbing athletes compared to active non-climbing athletes. Together with our findings in recreational climbers, the upper forearm musculature might have a greater beneficial effect in the climbing sport compared to other sport activities. However, other mechanisms apart from the measured circumference may influence strength as well, as none of the other circumference measures were different.

### 4.2. Grip Strength Assessment Serves as A Reliable Tool to Distinguish Climbers from Non-Climbers

We demonstrated that grip strength was superior in almost every but fist pinch in the recreational climbing cohort, however with mostly just moderate effect size. Using finger strength as parameter for climbing performance and to detect possible weaknesses in order to prevent fatigue-associated injuries is a valid tool among the climbing elite [33]. In order to measure grip strength, an arm fixation can produce more reliable results, but the free position as we used it is more related to natural climbing moves [33].

We hypothesized that the discrepancy in pinch force between climbers and non-climbers in both hands but non-dominant fist pinch might be due to the fact that this particular pinch is used for ordinary tasks more frequently, even in the non-climbing cohort, and subsequently better trained. By contrast, in climbing athletes, all other pinches are much better trained. This is why this effect remains consistent after adapting for BMI.

Watts and colleagues assessed grip strength using a study design similar to ours, however, with a focus on elite climbers. They did not detect differences in absolute hand strength between elite climbers and advanced climbers until adjusted to BMI. A difference to our measurement was that we used the mean value, while Watts et al. recorded the “highest reading”, which might have caused the missed difference [34].

With a grip dynamometer, Grant et al. demonstrated that elite climbers had higher grip strength (left, but not right) and pincer strength (I + II; left + right) compared to non- and recreational climbers. With a different apparatus for a hanging procedure, most grips were stronger in elite climbers compared to controls [17]. Consistent with findings of Grant et al., we measured that the “classical” pinch (I + II) on both sides was stronger in the climbing athlete compared to controls.

Azari et al. compared 24 male and female climbing athletes (IRCRA 21/12) finding a pincer strength (Pinch I/II) of 11.29 kg in male athletes and 7.08 kg in female athletes. Herein, the overall grip strength was 56.90 kg and 33.15 kg, respectively. All of these reached forces were higher compared to our findings. One reason could be that Azari et al. used a digital hand dynamometer (Saehan) and pincer dynamometer (Pinch Gauge) to measure grip strength, and the highest values were recorded [4]. On the other side, we measured the mean of two samples each. In summary, we were able to show that the differences in grip strength to other athletes that have been measured among the climbing elite are already in existence at a recreational level, but with a just moderate effect size. Since comparable studies demonstrated corresponding results for the climbing elite, and we measured enhanced grip strength in all but fist pinch, the small effect size indicates that recreational or intermediate climbers already show a small but measurable increase in force. Thus, not overall fist pinch force, but smaller pinches might be a promising target for beginning climbing athletes to improve on.

### 4.3. Differences between the Dominant and Non-Dominant Hand Are Pronounced in Non-Climbers

We found that differences between the dominant and non-dominant hand were more pronounced in non-climbers compared to climbers, however with a mostly small and trivial effect size. Notwithstanding, it might be beneficial for starting climbers, to be mindful that the grip strength training, which is advisable to practice (see Section Section 4.2), should be done on both hands simultaneously to overcome the force deficit in the non-dominant hand, which is prominent in non-climbers, but diminishes already in recreational climbers.

Handedness among climbers has been given little attention to, mostly in anecdotic reports [35]. Among professional climbers, differences between the dominant and non-dominant hand have been characterized on a physiological level. The oxygenation kinetics in M. flexor digitorum profundus in rock climbers differed significantly among the non-dominant and dominant hand, whereas the former was the most frequently involved in certain rock-climbing specific injuries [36]. These physiological insights cannot be directly transferred to grip force measurements, as performed by Grant et al., and a difference between both hands has not been insistently analyzed by this group. However, a diminishing discrepancy between both hands with rising climbing experience can be read out of their data [17]. We found that non-climbers depicted significant differences between the dominant and non-dominant hand in all pinches, whereas the climbers did not in Pinch I/IV and I/II+III+IV and could subsequentially demonstrate on a physiological level that grip strength differences between the dominant and non-dominant hand are more profound in the non-climbing cohort. Assumingly, unusual grips (like I/IV and I/II+III+IV) are more intensively trained by climbers, even in the non-dominant hand, resulting in an equalization of the differences between both hands.

### 4.4. Study Limitations

Since the overall study design is a cross-sectional study, no causative can be deduced. The results, however, need to be evaluated in randomized controlled studies. Hence, the associations and differences between climbers and athletes may be due to unmeasured confounders.

## 5. Conclusions

It has been demonstrated in previous studies, that the grip force of elite climbers exceeds the grip force of non-climbers by far. For the first time, we showed that these differences are prominent in recreational climbers as well. Subsequently, careful and gradual grip force training on both hands concurrently seems to be advisable for starters in the climbing sport.

The new finding in this study is that even in recreational climbing athletes, all pinch grips but non-dominant fist were superior compared to regular sportsmen. In addition, the grip force differences between the dominant and non-dominant hand appear larger in the non-climbing compared to the climbing cohort. The circumference 10 cm below the lateral epicondyle was significantly larger in the climbing cohort on both the dominant and non-dominant hand. As a consequence, athletes displaying both-handedness, large upper forearm circumference as well as strong single pinch force might be promising candidates for the climbing sport, whereas traditional athletes who want to start climbing might benefit from a training which focuses on the upper forearm and pinch grip strength on both hands.

## Figures and Tables

**Table 1 ijerph-18-00129-t001:** Analysis of circumference on defined locations between both groups. Means of climbers and non-climbers are compared to each other (unpaired t-test) and effect size (ES) is reported.

Location	Hand	Climbers			Non-Climbers			*p*	ES
		Mean	SD	95%-CI	Mean	SD	95%-CI		
15 cm above lateral epicondyle (cm)	Dom	28.82	2.871	28.0–29.6	29.08	3.039	28.2–29.9	n.s. (0.6662)	trivial
	Non-dom	28.65	2.836	27.9–29.4	28.74	3.289	27.8–29.7	n.s. (0.8768)	trivial
Greatest circumference upper arm (cm)	Dom	30.81	3.127	29.9–31.7	31	3.363	30.0–32.0	n.s. (0.7761)	trivial
	Non-dom	30.26	2.97	29.4–31.1	30.68	3.483	29.7–31.7	n.s. (0.5184)	trivial
Elbow (cm)	Dom	25.89	1.96	25.3–26.4	25.38	2.733	24.6–26.2	n.s. (0.2811)	trivial
	Non-dom	25.66	2.083	25.1–26.3	25.41	2.179	24.8–26.0	n.s. (0.5525)	trivial
Greatest circumference lower arm (cm)	Dom	26.9	2.374	26.2–27.6	26.17	2.293	25.5–26.8	n.s. (0.0761)	small
	Non-dom	26.57	2.392	25.9–27.2	25.83	2.366	25.2–26.5	n.s. (0.0997)	small
10 cm below lateral epicondyle (cm)	Dom	26.12	2.444	25.4–26.8	24.8	2.843	24.0–25.6	* (0.0142)	small
	Non-dom	25.68	2.617	24.9–26.4	24.55	2.511	23.8–25.3	* (0.0288)	small
Wrist (cm)	Dom	16.88	2.332	16.2–17.5	16.43	1.258	16.1–16.8	n.s. (0.5229)	trivial
	Non-dom	16.56	1.981	16.0–17.1	16.44	1.275	16.1–16.8	n.s. (0.8191)	trivial
Midhand (cm)	Dom	19.98	1.692	19.5–20.5	19.96	1.774	19.5–20.5	n.s. (0.9485)	trivial
	Non-dom	19.82	1.56	19.4–20.3	19.63	1.751	19.1–20.1	n.s. (0.5587)	trivial
Arm length (shoulder/distal ulna, cm)	Dom	59.45	3.807	58.4–60.5	60.09	3.601	59.1–61.1	n.s. (0.3914)	trivial
	Non-dom	59.47	3.812	58.4–60.5	60.11	3.634	59.1–61.1	n.s. (0.3905)	trivial
Hand span (greatest distance thumb to distal phalanx little finger, cm)	Dom	21.2	1.702	20.7–21.7	21.34	1.707	20.9–21.8	n.s. (0.6848)	trivial
	Non-dom	21.25	1.632	20.8–21.7	21.37	1.874	20.8–21.9	n.s. (0.7298)	trivial
Arm span (cm)	-	178.10	10.150	175.2–181.1	180.10	11.100	177.0–183.3	n.s. (0.3577)	trivial

* *p* < 0.05.

**Table 2 ijerph-18-00129-t002:** Analysis of different pinches with regards to grip strength between climbers and non-climbers measured in the dominant (Dom) and non-dominant (Non-dom) hand. Differences of each pinch between climbers and non-climbers are depicted (unpaired t-test), and effect size (ES) is reported.

	**Climbers**	**Non-Climbers**	***p*** **-Value**	**ES**
	**Dom**			
Pinch I/II (kg)	9.1 [8.4–9.8]	7.8 [7.1–8.4]	*** (0.0005)	moderate
Pinch I/III (kg)	9.3 [8.5–10.1]	8 [7.2–8.8]	** (0.0065)	small
Pinch I/IV (kg)	6.9 [6.1–7.6]	5.6 [5.0–6.2]	** (0.0011)	moderate
Pinch I/III+IV (kg)	11.7 [10.8–12.6]	10 [9.1–10.9]	** (0.0056)	moderate
Pinch I/II+III (kg)	13.5 [12.6–14.4]	11.7 [10.7–12.7]	* (0.0253)	moderate
Pinch I/II+III+IV (kg)	14.4 [13.3–15.5]	12.3 [11.3–13.2]	*** (0.0007)	moderate
Fist (kg)	44.2 [41.0–47.4]	41.0 [37.7–44.3]	* (0.0446)	small
	**Climbers**	**Non-Climbers**	***p*** **-Value**	**ES**
	**Non-dom**			
Pinch I/II (kg)	8.3 [7.6–9.0]	6.8 [6.2–7.3]	** (0.0058)	moderate
Pinch I/III (kg)	8.5 [7.7–9.3]	7.1 [6.4–7.8]	* (0.0243)	moderate
Pinch I/IV (kg)	6.6 [5.9–7.2]	5.2 [4.7–5.7]	** (0.0068)	moderate
Pinch I/III+IV (kg)	11 [10.0–11.9]	9.2 [8.3–10.0]	* (0.0105)	moderate
Pinch I/II+III (kg)	12.1 [11.2–13.0]	10.7 [9.9–11.6]	** (0.0072)	small
Pinch I/II+III+IV (kg)	13.9 [12.7–15.1]	11.4 [10.6–12.3]	** (0.0032)	moderate
Fist (kg)	42.0 [38.8–45.2]	37.3 [34.0–40.6]	n.s. (0.1650)	small

*** *p* < 0.001, ** *p* < 0.01, * *p* < 0.05.

**Table 3 ijerph-18-00129-t003:** Pinch force characteristics between the dominant (Dom) and non-dominant (Non-dom) hand among climbers and non-climbers. Differences are expressed between the right and left hand (paired t-test) and effect size is reported (ES).

	**Climbers**		***p*** **-Value**	**ES**
	**Dom**	**Non-dom**		
Pinch I/II (kg)	9.1 [8.4–9.8]	8.3 [7.6–9.0]	** (0.0013)	small
Pinch I/III (kg)	9.3 [8.5–10.1]	8.5 [7.7–9.3]	*** (<0.0001)	small
Pinch I/IV (kg)	6.9 [6.1–7.6]	6.6 [5.9–7.2]	n.s. (0.1071)	trivial
Pinch I/III + IV (kg)	11.7 [10.8–12.6]	11 [10.0–11.9]	** (0.0036)	trivial
Pinch I/II + III (kg)	13.5 [12.6–14.4]	12.1 [11.2–13.0]	*** (<0.0001)	small
Pinch I/II + III + IV (kg)	14.4 [13.3–15.5]	13.9 [12.7–15.1]	n.s. (0.0785)	trivial
Fist (kg)	44.2 [41.0–47.4]	42.0 [38.8–45.2]	** (0.004)	trivial
	**Non-Climbers**		***p*** **-value**	**ES**
	**Dom**	**Non-dom**		
Pinch I/II (kg)	7.8 [7.1–8.4]	6.8 [6.2–7.3]	*** (<0.0001)	small
Pinch I/III (kg)	8 [7.2–8.8]	7.1 [6.4–7.8]	*** (<0.0001)	small
Pinch I/IV (kg)	5.6 [5.0–6.2]	5.2 [4.7–5.7]	* (0.0101)	trivial
Pinch I/III + IV (kg)	10 [9.1–10.9]	9.2 [8.3–10.0]	*** (0.0002)	small
Pinch I/II + III (kg)	11.7 [10.7–12.7]	10.7 [9.9–11.6]	*** (<0.0001)	small
Pinch I/II + III + IV (kg)	12.3 [11.3–13.2]	11.4 [10.6–12.3]	*** (0.0007)	small
Fist (kg)	41.0 [37.7–44.3]	37.3 [34.0–40.6]	*** (<0.0001)	small

*** *p* < 0.001, ** *p* < 0.01, * *p* < 0.05.

## Data Availability

The data presented in this study are available on request from the corresponding author. The data are not publicly available due to restrictions of privacy.

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
