# Peer review of "Comparison of Grip Strength in Recreational Climbers and Non-Climbing Athletes—A Cross-Sectional Study"

_ijerph, 2020, doi:10.3390/ijerph18010129_

Round 1
Reviewer 1 Report
General comments:
I would like to acknowledge the commitment of the authors to follow the comments and suggestions made to improve the manuscript, which was definitely addressed and achieved. There are a few things that are still in need of improvement.
In general, I want to apologize for the comment on track changes. Apparently, the submitted manuscript already was a revised version (incl. track changes) which I was not aware of when I reviewed the manuscript.
Specific comments:
Title:
The title still includes „prospective cohort study“. Please change.
Abstract:
I still recommend to include actual p-values and effect sizes in the abstract (at least ranges) to provide the reader information on the magnitude of differences between recreational climbers and non-climbers.
Regarding the word „association“, I suggest to simply re-phrase the sentence: „Our results show/suggest that recreational climbers have…“.
Introduction:
LL41–43: I recommend to re-write this sentence. In its current form, one could infer that you compared elite climbers, recreational climbers, and recreational non-climbers in your study, which is not the case.
Discussion:
General: As you now included effect sizes in your results, I recommend to use them for interpreting the practical meaning of your results. There are some cases in which I would be more comfortable if you could moderate your language somewhat, e.g., the mostly trivial/small effect sizes between the dominant and non-dominant hand even in the non-climbers. Currently, you only state that there were significant differences between the dominant and non-dominant hand but do not acknowledge that these differences are only of trivial/small magnitude which might limit their practical relevance. Please also check for the remaining results.
LL220: I recommend to start this paragraph with a short introductory phrase to restate your own main findings as you did in the other paragraphs.
Conclusions:
General: I recommend to include practical applications derived from your study.
L240–241: In its current form, one could infer that you compared elite climbers, recreational climbers, and recreational non-climbers in your study, which is not the case. Please make clear that the findings regarding elite climbers were derived in previous studies, while you studied possible differences between recreational climbers and non-climbers (please see also comment on LL41–43).
Author Response
Reviewer #1:
General comments:
I would like to acknowledge the commitment of the authors to follow the comments and suggestions made to improve the manuscript, which was definitely addressed and achieved. There are a few things that are still in need of improvement.
In general, I want to apologize for the comment on track changes. Apparently, the submitted manuscript already was a revised version (incl. track changes) which I was not aware of when I reviewed the manuscript.
We thank the reviewer for the patience and further input to revise and improve the manuscript.
Specific comments:
Title:
The title still includes „prospective cohort study“. Please change.
We apologize for the mistake of not having the title of the manuscript changed when the title in the submission was. Accordingly, the title of the manuscript was changed into “Attributes of Recreational Climbers - A Cross-Sectional Study”.
Abstract:
I still recommend to include actual p-values and effect sizes in the abstract (at least ranges) to provide the reader information on the magnitude of differences between recreational climbers and non-climbers.
We thank for this helpful advice and added “(p<0.05 in all but non-dominant fist, small to moderate effect sizes), … (p=0.17), … (all p<0.05, all but one with small effect size), … (pinch I/IV and pinch I/II+III+IV not different and mostly trivial), … (p<0.05, small effect size).” to the abstract.
Regarding the word „association“, I suggest to simply re-phrase the sentence: „Our results show/suggest that recreational climbers have…“.
We thank the reviewer for this suggestion to simplify this sentence and changed it into “Our results show that recreational climbers have higher measured grip strength, but less profound differences between the dominant and non-dominant hand.”.
Introduction:
LL41–43: I recommend to re-write this sentence. In its current form, one could infer that you compared elite climbers, recreational climbers, and recreational non-climbers in your study, which is not the case.
We thank the reviewer and agree that the long nature of this sentence may be confusing. Subsequently, we shortened the sentence as following “Hence, whether also recreational climbing sportsmen are similarly superior compared to other athletic sportsman in terms of grip strength and anthropometric properties, has not been described in detail.”.
Discussion:
General: As you now included effect sizes in your results, I recommend to use them for interpreting the practical meaning of your results. There are some cases in which I would be more comfortable if you could moderate your language somewhat, e.g., the mostly trivial/small effect sizes between the dominant and non-dominant hand even in the non-climbers. Currently, you only state that there were significant differences between the dominant and non-dominant hand but do not acknowledge that these differences are only of trivial/small magnitude which might limit their practical relevance. Please also check for the remaining results.
We agree with the reviewer that previous language was very pronounced and changed this in the first paragraph with an additional “, the effect size was small [and]…”, in the second paragraph with “, however with mostly just moderate effect size.” and “… but with a just moderate effect size”, and in the third paragraph by adding “however with a mostly small and trivial effect size”.
LL220: I recommend to start this paragraph with a short introductory phrase to restate your own main findings as you did in the other paragraphs.
We thank the reviewer for this idea and start the paragraph now with an introductory sentence: “We found that differences between the dominant and non-dominant hand were more pronounced in non-climbers compared to climbers, …”, followed by a limiting “…however with a mostly small and trivial effect size”.
Conclusions:
General: I recommend to include practical applications derived from your study.
We thank the reviewer for pointing out the necessity of a practical deduction. As difficult as it is to conclude from our observations a practical applicability with not overstraining our (yet promising) results, we added “Athletes displaying both-handedness, large upper forearm circumference as well as strong single pinch force might be promising candidates for the climbing sport.” to the conclusions.
L240–241: In its current form, one could infer that you compared elite climbers, recreational climbers, and recreational non-climbers in your study, which is not the case. Please make clear that the findings regarding elite climbers were derived in previous studies, while you studied possible differences between recreational climbers and non-climbers (please see also comment on LL41–43).
We want to thank the reviewer again for these suggestions to improve this paragraph. We decided to now start with “It has been demonstrated in previous studies, that the grip force of elite climbers exceeds the grip force of non-climbers by far. For the first time, we showed that these differences are prominent in recreational climbers as well.”. For clarification, in the next sentence, the “not just in elite, but” has been deleted and the novelty of the subsequent finding emphasized.
Reviewer 2 Report
The article is relevant, since it differentiates between the dominant and the non-dominant member, beyond looking for differences between climbers and non-climbers, where the differences were predictable. In addition, they use a large sample size, allowing for more robust conclusions.
As for their format, on line 22, the dot "." after the word "measurements" should be removed.
Author Response
Reviewer #2:
The article is relevant, since it differentiates between the dominant and the non-dominant member, beyond looking for differences between climbers and non-climbers, where the differences were predictable. In addition, they use a large sample size, allowing for more robust conclusions.
We thank the reviewer for this positive feedback of our work.
As for their format, on line 22, the dot "." after the word "measurements" should be removed.
We thank the reviewer and removed the dot in line 22.
Reviewer 3 Report
This prospective/exploratory study aimed to identify indicators of climbing proficiency using upper body anthropometry and strength assessments, in recreational climbers. The findings are that expertise was associated with better grip strength and muscle circumference and a lesser effect of handedness on strength.
The main strength of this study was its methods, which were detailed. The size of the participant pool was also very interesting.
In spite of this, there were major disqualifying concerns with this study.
Firstly, the background does not cover the literature in nearly enough detail. I do not mean it should be necessary longer but the first paragraph is out of place. There's important scientific work missing (physical/physiological determinants of climbing proficiency).
Secondly, it is never clear what the actual aim of the study is. There's no actual research problem or knowledge gap exposed. Why should we know about recreational climber anthropometry or strength? Why these tests? The authors seem to hint at the Olympic scheduling but yet assess recreational climbers and non specialists.
Thirdly, there's an array of technical concerns. The sample size is never really stated (the reader gathers it's n=100 but there's no actual description of participant group). The results are exceedingly long for what are, in essence, simple results. Some attempt at brevity must be made here. The discussion skims the surface on the relevance of the work (see my first point, all problems start there IMO), and drawing at comparisons with few selected studies in the extent literature does not increase consistency unfortunately.
Overall, there are way too many issues to address in this work for it to be undertaken as part of the review process.
Author Response
Reviewer #3:
This prospective/exploratory study aimed to identify indicators of climbing proficiency using upper body anthropometry and strength assessments, in recreational climbers. The findings are that expertise was associated with better grip strength and muscle circumference and a lesser effect of handedness on strength.
We want to thank the reviewer for these comments, but think that we did not clarify the aim of our work sufficiently. We did not want to find indicators of climbing proficiency, but aimed to gather knowledge if the effects that have been seen in elite climbers (higher grip strength, certain anthropometric properties) can be similarly seen even in recreational athletes.
The interesting finding was, that if a similar active athletic group served as “control” (not a just “normal” control group without necessarily an athletic phenotype), just one anthropometric measurement indicated the climber. In addition, we discovered that not all pinches were stronger in climbers compared to other athletes, and that differences between both hands were less pronounced in recreational climbers compared to these athletes.
We apologize for the lack of explicitness in the introduction and added “Anticipating certain physical and physiological characteristics from previous studies of elite climbers [6–8,11–13], we sought to identify, whether these characteristics were already detectable in recreational climbers compared to similarly active sportsmen.”, aiming at a better clarification for the reason of our study.
The main strength of this study was its methods, which were detailed. The size of the participant pool was also very interesting.
In spite of this, there were major disqualifying concerns with this study.
Firstly, the background does not cover the literature in nearly enough detail. I do not mean it should be necessary longer but the first paragraph is out of place. There's important scientific work missing (physical/physiological determinants of climbing proficiency).
We thank the reviewer for these comments and ideas on how to improve the introduction. We changed the first paragraph of the introduction. In particular, and to not extend this part too much, we added important literature regarding insights into (elite) climbers’ physique: “A superior finger grip strength of the dominant hand in elite climbers has been plurally demonstrated, as has enhanced muscular endurance [5–7].”Additionally, we added the abovementioned sentence which indicates some of the excellent work on mostly elite climbers in the end of the Introduction.
Secondly, it is never clear what the actual aim of the study is. There's no actual research problem or knowledge gap exposed. Why should we know about recreational climber anthropometry or strength? Why these tests? The authors seem to hint at the Olympic scheduling but yet assess recreational climbers and non specialists.
We agree with the reviewer that this was not exhibited clearly before. The final sentence of the introduction now clarifies that elite climbers were described in detail before: “Anticipating certain physical and physiological characteristics from previous studies of elite climbers [6–8,11–13],…” and adds our research purpose in more clarity: “… we sought to identify, whether these characteristics were already detectable in recreational climbers compared to similarly active sportsmen.”.
Thirdly, there's an array of technical concerns. The sample size is never really stated (the reader gathers it's n=100 but there's no actual description of participant group).
We apologize for this mistake and agree with the reviewer that just mentioning the group size in the abstract and beginning of the discussion is not sufficient. Furthermore, we want to thank the reviewer for the courtesy regarding this aspect. We added “We included 50 climbing and 50 non-climbing athletes in this study.” to the 3.1 paragraph and the group size “n=50” to the Supplementary table 1 in both group headings.
The results are exceedingly long for what are, in essence, simple results. Some attempt at brevity must be made here.
We thank the reviewer for this proposal to improve the results. In accordance with that, we shortened the first paragraph, “Characteristics of Climbers and Non-Climbers” substantially, summarizing long descriptions into “Neither BMI nor gender or handedness differed significantly between the groups (Supplementary. Table 1).”.
The discussion skims the surface on the relevance of the work (see my first point, all problems start there IMO), and drawing at comparisons with few selected studies in the extent literature does not increase consistency unfortunately. Overall, there are way too many issues to address in this work for it to be undertaken as part of the review process.
We thank the reviewer for their opinion. After the purpose of our study has been more clearly stated in the introduction, we think that the discussion appears in a more favorable light now. Furthermore, we improved the discussion at its key point, the conclusion. In addition, we focused more on our own findings in the third paragraph “We found that differences between the dominant and non-dominant hand were more pronounced in non-climbers compared to climbers, however with a mostly small and trivial effect size.”. In order to highlight our results and integrate them into the existing knowledge, we added “It has been demonstrated in previous studies, that the grip force of elite climbers exceeds the grip force of non-climbers by far. For the first time, we showed that these differences are prominent in recreational climbers as well.” in the beginning of the Conclusions paragraph. To increase practical consistency, the final sentence deduces our findings into practical consequences: “Athletes displaying both-handedness, large upper forearm circumference as well as strong single pinch force might be promising candidates for the climbing sport.”.
We want to thank the reviewer for the critical review of our manuscript and the advice on how to improve it.
Round 2
Reviewer 1 Report
Thank you for your effort to further improve the manuscript. In my opinion, the manuscript can be accepted in its current form.
Author Response
We thank the reviewer for helping us to achieve these improvements.
Reviewer 3 Report
The very minor changes provided are very insufficient. My previous concerns regarding the paper and its issues remain.
Author Response
We regret that the revisions made have not been sufficient. As the editor-in-chief requested, we again changed the criticized parts substantially. We hope that these changes address the expressed concerns in a more persuasive manner now.
This manuscript is a resubmission of an earlier submission. The following is a list of the peer review reports and author responses from that submission.
Round 1
Reviewer 1 Report
I would like to appreciate the authors for their perfect responses to the required corrections.
No further comments.
Reviewer 2 Report
All of my previous concerns we completed in the new submission for this manuscript. Thank you.
Reviewer 3 Report
General comments:
Overall, I believe the authors have produced a study that provides novel information in the areas of rock climbing and recreational athletics. However, there are minor corrections I believe need to be addressed. If these corrections are made, I would suggest this paper be accepted.
Specific comments:
How were subjects recruited? Lines 56-58.
I believe it would be beneficial to have a reference on relationship between muscular size and strength at lines 80-81. Further, I would recommend some discussion on if size is necessary to influence strength, as the climbers appear stronger with only one significantly different circumference measure.
I would recommend reporting effect sizes.
Lines 184-187 – but these pinches are regularly conducted in the non-dominant hand of the climbers. I don’t know if this adequately explains the discrepancy in the non-dominant hand, as the dominant and non-dominant hand discrepancy was non-significant between climbers and non-climbers, as displayed in supplemental table 2. I would imagine this means that despite significant differences in the strength of both dominant and non-dominant hands between climbers and non-climbers, the discrepancy between non-dominant and dominant hands of climbers and non-climbers is similar. That being said, do climbers or non-climbers have a significant discrepancy between dominant and non-dominant hands? If there is, I believe this would speak volumes on the effect that daily activities of the dominant hand has, despite significant grip training through climbing.
Section 4.2 – grip performance was significantly better for climbers, while all but one circumference measure was not significantly different between groups. While volume or size of a muscle may influence strength, I think it should be noted that other mechanisms may influence strength, as none of the other circumference measures were different.
Reviewer 4 Report
General comments:
The authors have submitted a manuscript entitled „Attributes of Recreational Climbers - A prospective Cohort Study“. The study aimed to investigate differences in grip strength and arm circumference between recreational climbers and non-climbers. The information presented is interesting and could enhance our understanding of recreational climbers‘ physical attributes.
However, the manuscript has a number weaknesses that hamper the reading flow and the logical structure and should, therefore, be addressed. My major concerns about the manuscript are:
- The structure of the introduction and discussion section seem not logical to me
- The order in which the information is presented, is not consistent. For example, in the methods, you begin by describing the circumference measurements and end with the grip-force measurement. In the results section, this order is inverted.
- The discussion section heavily focuses on previous studies in this research field instead of focusing on your own results an putting them into the context of previous studies.
Moreover, the submitted manuscript seem to use track changes which makes it difficult for the reviewr to follow in some instances.
Following are some specific comments and suggestions to the authors to improve the quality of their manuscript.
Specific comments:
Title:
As to my knowledge, the term „prospective cohort study“ is not right here. In my opinion, your study design rather indicates a cross-sectional design.
Abstract:
- Why do you highlight the lack of studies on anthropometric characteristics but do not address this issue in your study? For clarity: I cannot find a clear statement in the manuscript that you measured upper extremity proportions as indicators of anthropometric characteristics.
- The results section should be re-written and actual p-values and effect sizes should be reported
- I would not use the word „association“ in this context as this usually refers to correlation analyses which you did not report in the results section.
- A discussion/conclusion section is currently missing and should, therefore, be included
Introduction:
Currently, the introduction is hard to follow and should be restructured. Moreover, the rationale of your study should be clearly worked out.
LL38–42: The whole paragraph is not clear to me. I do not understand the point you want to make with the first sentence. Also, I do not understand what the sentence on elite climbers adds to the reader.
L46: What information does this sentence add to the reader in the context of your study?
LL52–53: In which way does your study „guide the active athlete and give insights in what is possibly physiologically needed to join the climbing community“? Please clarify. This should also be addressed in the discussion section.
Materials & Methods:
General: Please be consistent with your wording throughout the manuscript (e.g., grip strength vs. grip force).
L56: As to my knowledge, the term „prospective cohort study“ is not right here. In my opinion, your study design rather indicates a cross-sectional design.
L75: This information has already been presented in L65 and, thus, can be deleted.
L80: „To indirectly assess muscular strength…“ This statement needs to be supported by literature.
LL88–89: The reason for grip force being measured does not need to be re-stated in this section.
LL106–109: Why is this result presented in the methods section? Moreover, you do not refer to this result in the discussion section.
Results:
General: I strongly recommend to insert effect sizes to quantify the magnitude of differences between climbers and non-climbers.
Discussion:
General: I recommend to restate your main findings in the first paragraph of the discussion section.
Currently, the discussion section is hard to follow. While the sub-headings improve the structure of this section, the paragraphs itselves (4.1 and 4.2) seem rather unstructured to me.
Moreover, the discussion section heavily focuses on previous studies in this research field instead of focusing on your own results an putting them into the context of previous studies.
LL157–158: You mention „similar studies“ but cite only one study. Please clarify.
LL167–168: „… higher in men…“ Compared to?
LL182–183: In contrast to this assumption, the test reliability was extremely high (ICC = 0.99). This leads one to conclude that there were no big differences between the best and the worst trial. Please clarify.
L173–174: It is not clear to me whether you are referring to your own study or to [21].
LL184–185: I do not get to which results of Table 1 you are referring to. Please clarify.
LL194–196: Please provide a reference for the respective study.
L196: „Consistently“ This word is misleading in my opinion, as you state the opposite of the study above. Please consider replacing by „In contrast“.
L214: Please see comment on „prospective“.
L214: Why are your results just descriptive? Please clarify in the light of you using inferential statistics such as t-tests.
LL217–218: I do not think that your sample size is a limitation as your power analysis revealed a necessary sample of 42 participants.
Conclusions:
L223: What do you mean by „… as observed in elite climbers.“? Please clarify.
LL223–225: This sentence is not clear to me, pease re-write.